# Analysis of High-Speed Milling Surface Topography and Prediction of Wear Resistance

**DOI:** 10.3390/ma15051707

**Published:** 2022-02-24

**Authors:** Wei Zhang, Kangning Li, Weiran Wang, Ben Wang, Lei Zhang

**Affiliations:** 1Key Laboratory of Advanced Manufacturing and Intelligent Technology, Ministry of Education, Harbin University of Science and Technology, Harbin 150080, China; likangning9710@163.com (K.L.); libaozhi9710@163.com (W.W.); lyx13667831689@163.com (B.W.); leizhang202201@163.com (L.Z.); 2College of Mechanical and Power Engineering, Harbin University of Science and Technology, Harbin 150080, China

**Keywords:** high-speed milling, topography parameters, BP neural network, prediction of wear resistance

## Abstract

Surface topography parameters are an important factor affecting the wear resistance of parts, and topography parameters are affected by process parameters in order to explore the influence law of process parameters on surface topography parameters and to find the quantitative relationship between milling surface topography parameters and wear resistance. Firstly, this paper took the surface after high-speed milling as the research object, established the residual height model of the milled surface based on static machining parameters, and analyzed the relationship between the residual height of the surface and the machining parameters. Secondly, a high-speed milling experiment was designed to explore the influence law of processing parameters on surface topography and analyzed the influence law of processing parameters on specific topography parameters; Finally, a friction and wear experiment was designed. Based on the BP neural network, the wear resistance of the milled surface in terms of wear amount and friction coefficient was predicted. Through experimental verification, the maximum error of the prediction model was 16.39%, and the minimum was 6.18%.

## 1. Introduction

Surface wear is the main factor affecting the service performance of parts, and how to improve the wear resistance of the surface of the parts has always been a research hotspot in the manufacturing industry. 

In recent years, some scholars have found through research that good wear resistance can be obtained when the surface of the part has some special topography [1,2,3]. Braun et al. used the dial test method to characterize the friction and wear behavior of steel sliding pairs with a diameter of 15–800 µm under mixed lubrication conditions, stated that at a certain sliding speed, using the best texture diameter can reduce friction by 80% and reduce wear [4]; Tillmann W et al. used the micro-milling method to prepare surfaces with honeycomb and dimple topographies using high-speed steel materials, and focused on the analysis of the impact of the surface topography on its friction and wear properties [5]; Conradi et al. studied the different morphologies of the Ti6Al4V surface: linear, cross-corrosion and dimples, and analyzed the effect of titanium alloy surface topography, weave density, and orientation (parallel, perpendicular and at 45°) on frictional wear under dry and lubricated conditions [6]; Razfa et al. pointed out that the micro topography of the machined surface has a great influence on the surface wear performance of the parts, and the analysis and research of the surface topography is of great significance to the wear resistance of the product [7]; Wiciak-Pikuła used machine learning algorithms to predict the surface morphology parameters Ra and Rz of composite materials, and the effectiveness of the prediction model was verified by experiments [8]; Feng et al. found that the fabric surface with high entanglement and nano-structured particles could obtain high abrasion resistance [9]; Daymi et al. designed an experiment for ball-end milling Ti-6Al-4V titanium alloy, and explored the influence law of the machining inclination on the milling surface at the same time, a prediction model of milled surface roughness with cutting speed, feed, and radial depth of cut as variables was also established [10]; Taoheed et al. studied the influence of different spindle speeds and feed rates on the surface topography of aluminum-based alloys. The test results showed that the surface roughness will decrease as the spindle speed decreases [11]; Mardi et al. studied the influence of kinematic parameters on the surface morphology of nanocomposites, and expressed the results through topography parameters [12]; Maher et al. established an adaptive neuro-fuzzy system of machining parameters and surface roughness by studying and analyzing the correlation between machining parameters (spindle speed, feed per tooth, depth of cut), milling forces, and surface roughness [13]; Vishwas et al. investigated the effect of process parameters such as cutting speed, feed, and depth of cut on the surface topography of martensitic stainless steel by means of turning machining [14].

The above research provides a new idea for improving the wear resistance of parts. The purpose of improving wear resistance can be achieved by preparing some special topographies on the surface of the parts; therefore, the purpose of preparing ideal surface topography can be achieved by exploring the influence of processing parameters on surface topography. Some studies have shown that the topography parameter is an intuitive manifestation of the surface topography and can be used to characterize the surface topography [15,16,17,18]; therefore, the influence of the processing parameters on the surface topography can be further explored by analyzing the relationship between the topography parameters and the processing parameters. In addition, there is also a certain relationship between the topographic parameters and the wear resistance of the parts [19,20,21]. The relationship between the topography parameters and the wear resistance of the parts can be used to establish a wear resistance prediction model to predict the wear resistance of the parts.

Regarding the prediction of part wear resistance, Wang et al. established a zero-order six-variable gray model for predicting wear characteristic parameters using finite element simulation technology and gray relational analysis [22]; Zhang et al. proposed a method to predict the amount of wear by using the size of the surface topography of parts [23]; Durmuş et al. used artificial neural networks to predict the amount of wear on 6315 aluminum alloy under different conditions [24]; Mahdi proposed a novel hybrid machine learning method for tool wear prediction based on XGBoost-SDA, and used simulation to verify the effectiveness of the prediction method [25]; Suresh et al. used the response surface method to minimize the test conditions, established a mathematical model of the wear rate, and predicted the wear rate on a 99.5% confidence interval [26]; Dursun et al. established an artificial network prediction model to predict the wear of A356 composite material. Compared with the test data, the correlation coefficient R2 is 0.9855, and the prediction model has a high degree of credibility [27]; Zhao et al. proposed a numerical method for joint wear prediction of rotational gap joints in a flexible mechanical system that combines wear prediction with flexible multi-body dynamics [28].

Based on previous research, a model of the residual height of the milled surface was established, the effect of processing parameters on surface topography and specific topographic parameters was analyzed, and the dual-indicator wear resistance prediction was completed based on BP neural network for specific topographical parameters.

## 2. Analytical Modeling of Residual Height of Ball-End Milling Surface

After the workpiece is processed by high-speed milling, a part of the material will remain on the surface. Its manifestation is the pit topography surrounded by four surface ridgelines, and its corresponding height value is called the residual height of the surface. The residual height of the surface is an important index to evaluate the surface micro-topography; therefore, a model of the residual height of the surface was established in the feed direction and the row spacing direction, as shown in Figure 1 and Figure 2. Among them, *f_z_* represents the feed per tooth, and *a_e_* represents the row spacing.

### 2.1. Residual Height Modeling in Feed Direction

The projection view of the characteristic ridgeline of the feed direction of the surface topography micro-unit in the *xoz* plane is shown in Figure 1. *V*_1_ and *V*_2_ are the projections of the ridgelines corresponding to the highest point and the lowest point of the topographic unit on the *xoz* plane.

The mathematical relationship between the arc radius *r*′ of the ridgeline *V*_1_ and the arc radius *r* of the ball end mill exists in the following equation:(1)$$r^{\prime }=\sqrt{r^{2}-{\left(f_{z}/2\right)}^{2}}$$

*H*_max_ is the maximum residual height on the surface topography unit, which represents the distance from the highest point to the lowest point of the topography unit. The calculation equation of *H*_max_ is as follows:(2)$$H_{max}=r-\sqrt{r^{\prime 2}-{\left(a_{e}/2\right)}^{2}}$$

A point Q was chosen arbitrarily on the ridgeline *V*_1_, and *i* is the distance between the projection of point Q on the *x*-axis and point O. Since the projection plane of the ridgeline *V*_1_ is perpendicular to the feed direction, *H*_1_ represents the height from point Q to the lowest point of the micro-unit profile. The calculation equation of *H*_1_ is as follows:(3)$$H_{1}=r-\sqrt{r^{\prime 2}-i^{2}}(0\leq i\leq a_{e}/2)$$

The calculation equation of the height *H*_2_ of the point on the ridgeline *V*_2_ corresponding to the point Q selected by the ridgeline *V*_1_ is as follows:(4)$$H_{2}=r-\sqrt{r^{2}-i^{2}}(0\leq i\leq a_{e}/2)$$

According to Equations (3) and (4), the residual height ridgeline *V*_1_ and *V*_2_ trajectory matrix expressions of the surface topography micro-unit projected in the feed direction can be obtained as follows:(5)$$V_{1}={\left[\begin{array}{ccc} x_{1} & y_{1} & z_{1} \end{array}\right]}^{T}={\left[\begin{array}{ccc} i & f_{z} & H_{1} \end{array}\right]}^{T}={\left[\begin{array}{ccc} i & f_{z} & r-\sqrt{r^{\prime 2}-i^{2}} \end{array}\right]}^{T}$$
(6)$$V_{2}={\left[\begin{array}{ccc} x_{2} & y_{2} & z_{2} \end{array}\right]}^{T}={\left[\begin{array}{ccc} i & f_{z} & H_{2} \end{array}\right]}^{T}={\left[\begin{array}{ccc} i & f_{z} & r-\sqrt{r^{2}-i^{2}} \end{array}\right]}^{T}$$

### 2.2. Residual Height Modeling in Row Spacing Direction

The projection view of the characteristic ridgeline of the row spacing direction of the surface topography micro-unit in the *yoz* plane is shown in Figure 2. *V*_3_ and *V*_4_ are the projections of the ridgelines corresponding to the highest point and the lowest point of the topographic unit on the *yoz* plane.

The mathematical relationship between the arc radius *r*″ of the ridgeline V_3_ and the arc radius *r* of the ball end mill exists in the following equation:(7)$$r^{\prime \prime }=\sqrt{r^{2}-{\left(a_{e}/2\right)}^{2}}$$

*H*_max_ is the maximum residual height on the surface topography unit, which represents the distance from the highest point to the lowest point of the topography unit. The calculation equation of *H*_max_ is as follows:(8)$$H_{max}=r-\sqrt{r^{\prime \prime 2}-{\left(f_{z}/2\right)}^{2}}$$

A point K was chosen arbitrarily on the ridgeline *V*_3_, and *j* is the distance between the projection of point K on the *y*-axis and point O. The calculation equation of *H*_3_ is as follows:(9)$$H_{3}=r-\sqrt{r^{\prime \prime 2}-j^{2}}(0\leq j\leq f/2)$$

The calculation equation of the height *H*_4_ of the point on the ridgeline *V*_4_ corresponding to the point K selected by the ridgeline *V*_3_ is as follows:(10)$$H_{4}=r-\sqrt{r^{2}-j^{2}}(0\leq j\leq f/2)$$

According to Equations (9) and (10), the residual height ridgeline *V*_3_ and *V*_4_ trajectory matrix expressions of the surface topography micro-unit projected in the row spacing direction can be obtained as follows:(11)$$V_{3}={\left[\begin{array}{ccc} x_{3} & y_{3} & z_{3} \end{array}\right]}^{T}={\left[\begin{array}{ccc} a_{e} & j & H_{3} \end{array}\right]}^{T}={\left[\begin{array}{ccc} a_{e} & j & r-\sqrt{r^{\prime \prime 2}-j^{2}} \end{array}\right]}^{T}$$
(12)$$V_{4}={\left[\begin{array}{ccc} x_{4} & y_{4} & z_{4} \end{array}\right]}^{T}={\left[\begin{array}{ccc} a_{e} & j & H_{4} \end{array}\right]}^{T}={\left[\begin{array}{ccc} a_{e} & j & r-\sqrt{r^{2}-j^{2}} \end{array}\right]}^{T}$$

In summary, based on the ridgeline expression of the residual height in the feed direction and the line spacing direction, the residual height matrix of the surface topography obtained by ball-end milling can be expressed as:(13)$$V={\left[\begin{array}{cc} V_{3} & V_{1} \end{array}\right]}^{T}$$

It is precisely because of the periodic arrangement of the residual height ridges of the two types of feed direction and line spacing direction of *V*_1_ and *V*_3_ that the surface topography obtained by high-speed ball milling shows a regular pit shape.

Combining Equations (1),(2),(7) and (8), we can obtain the relationship between the maximum residual height and the processing parameters as follows:(14)$$H_{max}=r-\sqrt{r^{2}-(f_{z}{}^{2}+a_{e}^{2})/4}$$

It can be seen from Equation (14) that the maximum residual height of the surface is affected by the radius of the ball end milling cutter, *f_z_* and *a_e_*, and it is positively correlated with *f_z_* and *a_e_*.

## 3. High-Speed Milling Experiment and Topography Detection

### 3.1. Experimental Equipment and Specimen Materials

As shown in Figure 3, the machine tool used in this milling experiment is the DMU 60 monoBLOCK five-axis vertical machining center(DMG company, Bielefeld, Germany). An indexable insert ball-end milling cutter was used. The blade model is BNM200-TG. The blade diameter is 20 ± 0.006 mm. The base material of the blade is cemented carbide. The surface is coated with JC6102, and the hardness is about 70 HRC. The cutting edge line is “S” type. The material selected for the experiment is heat-treated Cr12MoV die steel, and its main chemical composition is shown in Table 1. The inclination angle of the ball-end milling cutter during milling is controlled by the machining center. High-speed cutting was used in the milling experiment.

The experimental testing equipment used the ultra-depth-of-field microscope (KEYENCE company, Osaka, Japan) and Taylor Map CCI white light interferometer (Taylor Hobson company, Leicester, UK). Figure 4a is the ultra-depth-of-field microscope, and Figure 4b is Taylor Map CCI white light interferometer.

### 3.2. Experimental Parameter Design

According to the analysis of the residual height of the milling surface in the first section, in order to explore the relationship between the main processing parameters and the surface topography, the following single-factor experimental schemes for different processing parameters are designed, *a_e_*, *a_p_*, and *f_z_* were analyzed in Table 2, Table 3, and Table 4 using the control variable method, respectively.

### 3.3. Milling Topography Detection and Analysis

The surface detection of the milled workpiece is carried out with the ultra-depth-of-field microscope, and the detection results are shown in Figure 5, Figure 6 and Figure 7.

It can be seen from Figure 5 that the overall size of the scallop crater topography tends to increase as *a_e_* increases. This is because as *a_e_* increases, the distance between the ridgelines of the texture of two adjacent rows increases. Comparing the effects of different *a_p_* in Figure 6, it is found that the overall size of the scallop pits does not change significantly. The reason for this phenomenon is that the smaller *a_p_* causes less material residue on the surface, making the influence of *a_p_* insignificant. As can be seen from Figure 7, when *f_z_* gradually increases from 0.5 to 0.9, the micro-texture topography of the processed surface begins to be disordered. The main reason is that the cutting force is too large, which causes the “wrong tool” phenomenon caused by the vibration of the tool and the machine tool.

### 3.4. Topography Parameter Detection

The white light interferometer is used to detect the shape parameters of the milled workpiece. The results are shown in Table 5.

## 4. Analysis of Milling Surface Topography Parameters

### 4.1. Characterization of Milling Topography Parameters

#### 4.1.1. Parameter Characterization of Pit Topography in Vertical Direction

The maximum residual height *H*_max_ of the pit topography of the ball-end milling surface is the distance from the lowest point of the pit topography to the highest point of the remaining topography. The shape parameter *S_z_* expresses the sum of the maximum peak height *S_p_* and the maximum valley depth *S_q_* in the evaluation area. Since the inclination angle of this processing is 30°, combining Equation (14), the equation between *S_z_* and *H*_max_ is as follows:(15)$$H_{max}=\sqrt{3}(r-\sqrt{r^{2}-(f_{z}^{2}+a_{e}^{2})/4})/2=S_{z}=S_{p}+S_{v}$$

By substituting the radius r of the ball end mill, *a_e_* and *f_z_* and comparing them with the measured *S_z_*, they are approximately equal; therefore, *S_z_*, *S_p_*, *S_v_* in the topography parameters can be used to characterize the maximum residual height of the surface topography of the ball-end milling pit, which is the size of the pit topography in the vertical direction.

The height ratio of surface features *S_tr_* is a parameter for judging whether the surface topography has directionality or not. When *S_tr_* is close to 0, the topography is directional, as shown in Figure 8a. When it is close to 1, the topography does not depend on the direction, as shown in Figure 8b.

The greater the difference between the height of the residual ridgeline in the direction of the row spacing and the height of the residual ridgeline in the feed direction, the stronger the directionality of the topography; the smaller the difference, the weaker the directionality; therefore, *S_tr_* can be used to characterize the size relationship of the edge height dimension of the pit topography between the feed direction and the row spacing direction.

#### 4.1.2. Parameter Characterization of Pit Topography in Horizontal Direction

When the height distribution between the topography is similar, the minimum autocorrelation length *S_al_* represents the distance between two similar height points. Since the height distribution of each group of ball-head milling dimple profiles is similar, *S_al_* can be used to represent the distance between the maximum residual height points of two adjacent pits; therefore, the minimum autocorrelation length *S_al_* can be used to characterize the overall size of the pit topography in the horizontal direction.

The interface expansion area ratio *S_dr_* represents the increased value of the actual topographic surface area relative to the reference plane. When there are more pit topographies in a certain area, that is, the overall size of the pit topography micro-unit is smaller, the interface expansion area ratio *S_dr_* will be larger; conversely, the larger the overall size of the pit shape micro-unit, the smaller the Sdr; therefore, the interface expansion area ratio *S_dr_* can also be used to characterize the overall size of the pit topography in the horizontal direction.

In the combined analysis above, *S_z_*, *S_p_*, *S_v,_* and *S_tr_* are used to comprehensively characterize the size of the vertical height direction of the pit topography of the ball milling surface; *S_al_* and *S_dr_* are used to comprehensively characterize the size of the horizontal direction of the pit topography of the ball milling surface.

### 4.2. Analysis of the Influence Law of Processing Parameters on Topography Parameters

According to Table 2 and Table 5, the changes of the topography parameters under the change of *a_e_* are shown in Figure 9.

It can be seen from Figure 9 that with the increase in *a_e_*, the topography parameter values of *S_z_*, *S_p_*, *S_v,_* and *S_tr_* are also increasing, while *S_al_* and *S_dr_* are gradually decreasing trend. Since *S_z_*, *S_p_*, and *S_v_* are all different expressions of the height distribution of surface topography, as *a_e_* increases, the maximum residual height will increase, so the *S_z_*, *S_p_*, and *S_v_* parameter values increase. In addition, the increase in *a_e_* will increase the height of the residual ridgeline in the feed direction, making it close to the height of the residual ridgeline in the row spacing direction. As the heights of the remaining ridges in the two directions approach, the directionality of the topography becomes less obvious, which then shows an increase in the *S_tr_* value. When *a_e_* increases, the amount of ball-end milling cutter cutting into the workpiece increases, resulting in an increase in the overall size of the pit topography. The distance between adjacent points with similar heights will increase, which then shows a decrease in the *S_al_* value. As the overall size of the pits becomes larger, the number of pits in the same area will decrease, which then shows a decrease in the *S_dr_* value.

According to Table 3 and Table 5, the changes of the topography parameters under the change of *a_p_* are shown in Figure 10.

It can be seen from Figure 10 that with the increase in *a_p_*, there is no obvious trend change in the changes of various shape parameter values. This is because the increase in *a_p_* fails to significantly change the maximum residual height value, so the changes in various topography parameters are not obvious.

According to Table 4 and Table 5, the changes of the topography parameters under the change of *a_p_* are shown in Figure 11.

With the increase in *f_z_*, the topography parameter values of *S_z_*, *S_p_*, *S_v_,* and *S_tr_* also increase, while *S_al_* and *S_dr_* gradually decrease. This changing trend is consistent with the influence of *a_e_* on topographical parameters. The main reason is that the values of *f_z_* and *a_e_* have a positive correlation between the maximum residual height of the topography and the overall size of the pit topography unit.

## 5. Prediction and Verification of Milling Surface Wear Resistance

### 5.1. Friction and Wear Experiment

#### 5.1.1. Experimental Equipment and Sample Preparation

The friction and wear experiment adopts the MFT-5000 multifunctional friction and wear tester produced by Rtec Corporation of the United States.

Figure 12 shows the friction and wear tester, the wire-cutting machine was used to cut the test piece completed in the milling test from the test material. The size of the specimen used for the friction and wear test is 30 × mm × 16 mm × 6 mm. The upper sample size is 10 mm × 5 mm × 3 mm, and material selection is YS8.

The machining parameters of the last four groups of milling tests are selected relatively large so that the cutting force is too large, and the vibration of the machine tool and the tool is more obvious. This leads to an increase in the complexity of the surface topography, so the measured topography parameters cannot effectively restore the forming mechanism; therefore, only the first nine groups of samples after the milling test were selected for the friction and wear test.

The friction and wear experimental parameters in this section are as follows: the sliding frequency is 2 Hz, the single stroke is 20 mm, the applied load is 80 N, and the test time for each group of specimens is 6 min.

#### 5.1.2. Topography Detection after Wear

The surface detection of the workpiece after the friction and wear experiment was carried out with the ultra-depth-of-field microscope, and the detection results are shown in Figure 13.

#### 5.1.3. Wear Data Detection

Before the friction and wear test, the test sample was cleaned with an ethanol solution, dried with a hairdryer, and weighed with an electronic balance (minimum measurement value 0.1 mg). After the friction and wear test was completed, the test sample was cleaned with an ethanol solution, dried with a hairdryer, and then weighed on an electronic balance. We record the mass *m*_1_ of the test specimens before the test and *m*_2_ of the test specimens after the test, and calculate the wear of each group of test specimens.

The calculation equation for the amount of wear is as follows:(16)$$\Delta m=m_{1}-m_{2}$$

The friction coefficient is measured by the detection software of the friction and wear test equipment. Since the friction coefficient is a real-time monitoring measurement value, there may be a large measurement error immediately before the end of the friction and wear test. Here we stipulate that the real-time friction coefficient value at 359.5 s is selected. The measurement data are summarized as shown in Table 6.

### 5.2. Wear Resistance Evaluation Index Prediction and Verification

The relationship between milling surface topography and wear and the friction coefficient is very complicated. From the previous analysis, it is known that *S_p_*, *S_z_*, *S_dr_*, *S_al_,* and *S_v_* can provide a relatively complete characterization of the milled topography; therefore, the relationship between multiple topography parameters and wear resistance indexes is comprehensively analyzed, and the wear resistance prediction model is constructed using BP neural network.

#### 5.2.1. BP Neural Network Parameter Selection

Input layer and output layer design

The number of nodes in the input layer is the dimensionality of the selected training sample, and the dimensionality of the predictor index is the number of nodes in the output layer. This article studies the relationship between the milling surface topography parameters analyzed in the previous chapter (*S_p_*, *S_z_*, *S_dr_*, *S_al_,* and *S_v_*) and surface wear resistance evaluation indicators; therefore, the input layer is the five topography parameters, and the output layer is the amount of wear (the method of predicting the coefficient of friction is consistent with it). Finally, it is determined that the number of nodes in the input layer of the prediction model is 5, and the number of nodes in the output layer is 1.

2.Hidden layer design

The calculation equation of the hidden layer is as follows:(17)$$i=\sqrt{k+l}+c$$

In the equation, *i* is the number of hidden layer nodes; *k* and *l* are the number of input layer and output layer nodes; *c* is a constant between 1 and 10.

Substituting the number of input layer nodes and the number of output layer nodes into Equation (17), the interval of the number of hidden layer nodes is [4,12]. In order to obtain the optimal value of the number of hidden layer nodes, each value of the interval is substituted into the MATLAB prediction program to calculate. The model error obtained is shown in Figure 14.

From the analysis of Figure 14, the number of hidden layer nodes is 12 when the model error is minimal.

Therefore, the number of nodes in the input layer, hidden layer, and output layer can be finally determined to be 5, 12, and 1. The structure of the BP neural network prediction model is shown in Figure 15.

#### 5.2.2. Training Sample Selection

This article selects A1, A4, and B4 in Table 7 as test samples and the remaining six groups as training samples.

#### 5.2.3. Sample Normalization

Neural network normalization processing is realized by correlation function. This article selects the most commonly used mapminmax function. The normalization equation of mapminmax function is as follows:(18)$${x_{i}}^{\prime }=\frac{x_{i}-x_{min}}{x_{max}-x_{min}}$$

In the equation, *x_i_* and *x_i_*’ are the sample parameter values before and after normalization, respectively; *x*_max_ and *x*_min_ are the maximum and minimum values of each index data, respectively. The data after normalization are shown in Table 8.

#### 5.2.4. Wear Resistance Prediction

The training function and the transfer function of the prediction model have respectively selected the trading function and the purelm function. The learning efficiency and learning step length are, respectively, 0.05 and 0.7. The minimum mean square error was set to 0.0001. The maximum number of training sessions was set to 100. After continuous optimization, the BP neural network prediction model was completed, and the prediction program was written using MATLAB.

The wear prediction program is used to predict the wear. After 14 sample training, the error dropped to the lowest value 3.7582 × 10^−7^. The error curve is shown in Figure 16.

#### 5.2.5. Validation of Wear Resistance Prediction Model

We input the training samples into the model, and continuously carried out the error reverse transmission correction. When the error value reached the specified value, the training ended; that is, the model completed learning. At this point, the three sets of test samples set were input into the model, and the prediction of wear amount began. Finally, the output predicted values were compared with the experimental measurements for error calculation. The same method was used to complete the friction coefficient prediction. The calculation results are shown in Table 9 and Table 10.

The maximum relative error of the predicted values of each group is 16.39%, and the minimum is 6.18%. It can be explained that the wear resistance prediction model established is relatively accurate in predicting the evaluation indicators of the milling surface wear resistance. In addition, due to the positive correlation between the amount of wear and the coefficient of friction, the correctness of the wear resistance prediction model can also be proved from the side.

## 6. Conclusions

This paper takes the surface topography of Cr12MoV die steel after high-speed milling as the research object, the relationship between the residual height of the surface and the processing parameters was studied, the influence law of processing parameters on specific topography parameters was analyzed, and the wear resistance is predicted based on the topography parameters. The study’s general conclusions are as follows:The model of the residual height of the ball-end milling surface was established, and the relationship between the residual height of the surface and the processed static parameters was obtained. The residual height value of the processed surface is determined by the size of the ball end mill, *f**_z_* and *a_e_*, and has a positive correlation with *f**_z_* and *a_e_*.Different processing parameters will have different effects on surface topography. The increase in *a_e_* will make the surface topography unit larger, and the change of *f_z_* will make significant changes in the residual height of the surface topography.There is an influential relationship between processing parameters and topography parameters. There is a positive correlation between *S_z_*, *S_p_*, *S_v_*, *S_tr_,* and *a_e_* and *f_z_*; There is a negative correlation between *S_al_*, *S_dr_*, and *a_e_* and *f_z_*.A wear resistance prediction model based on topographical parameters was developed using BP neural network, and the prediction of the wear and friction coefficient of the ball-end milled surface is accomplished by inputting topographical parameters. The maximum relative error of the predicted value is 16.39%, and the minimum is 6.18%.

## Figures and Tables

**Figure 1 materials-15-01707-f001:**
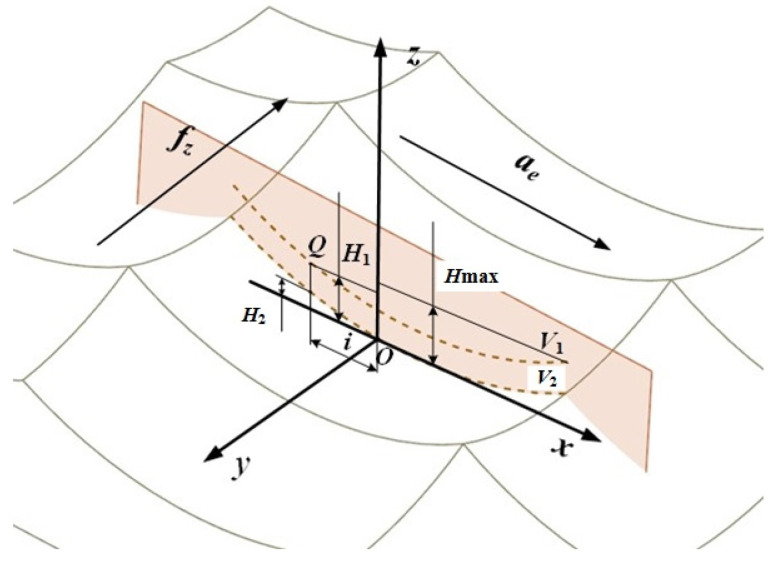
Feed direction characteristic ridgeline.

**Figure 2 materials-15-01707-f002:**
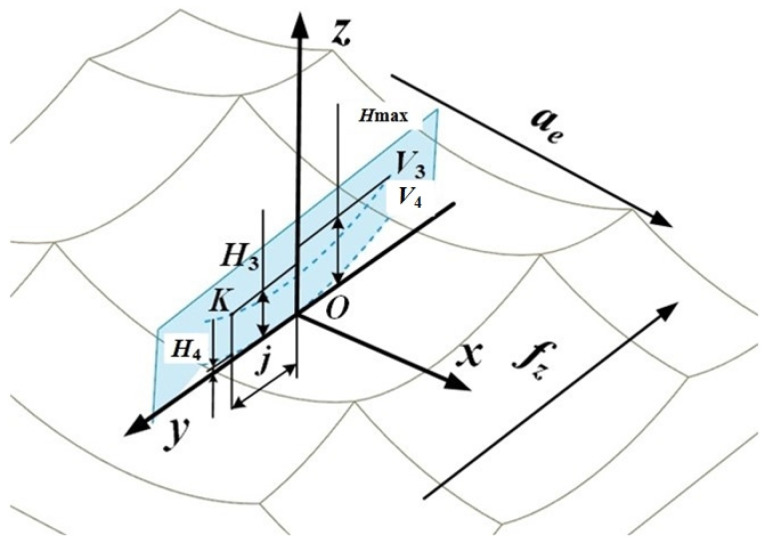
Row spacing direction characteristic ridgeline.

**Figure 3 materials-15-01707-f003:**
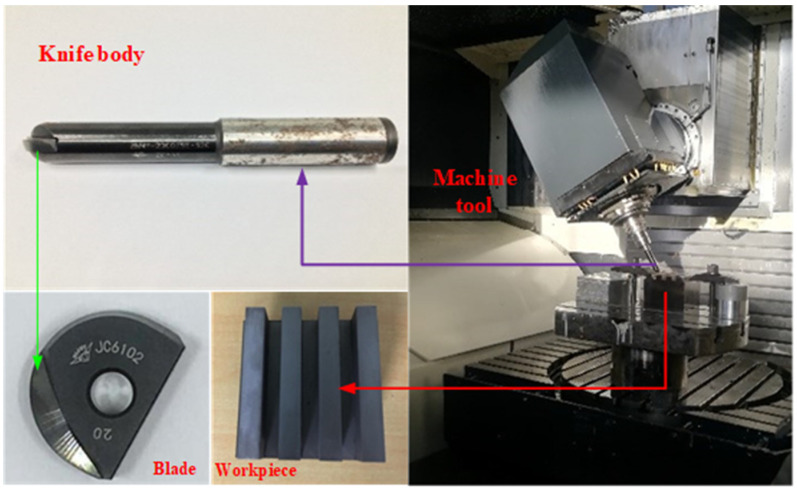
Machining center and milling experiment site.

**Figure 4 materials-15-01707-f004:**
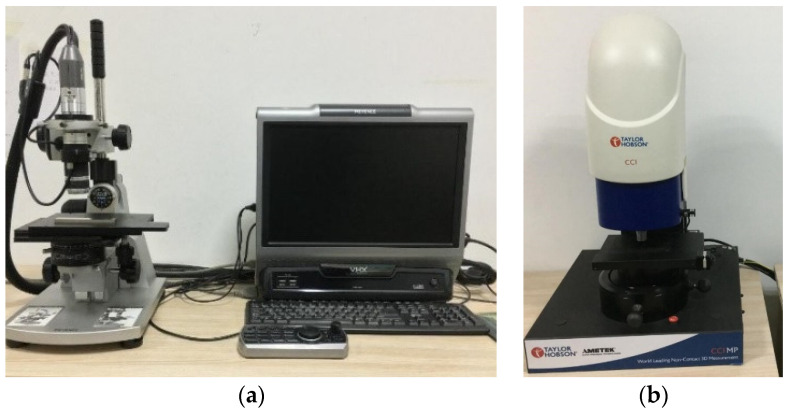
The experimental testing equipment: (**a**) the ultra-depth-of-field microscope; (**b**) the white light interferometer.

**Figure 5 materials-15-01707-f005:**
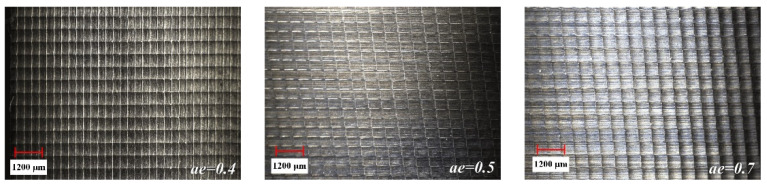
The influence of *a_e_* change on topography.

**Figure 6 materials-15-01707-f006:**
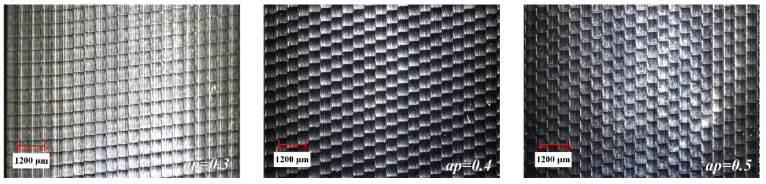
The influence of *a_p_* change on topography.

**Figure 7 materials-15-01707-f007:**
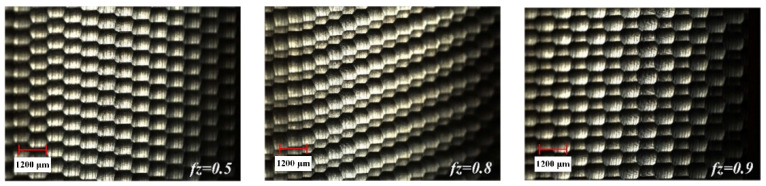
The influence of *f_z_* change on topography.

**Figure 8 materials-15-01707-f008:**
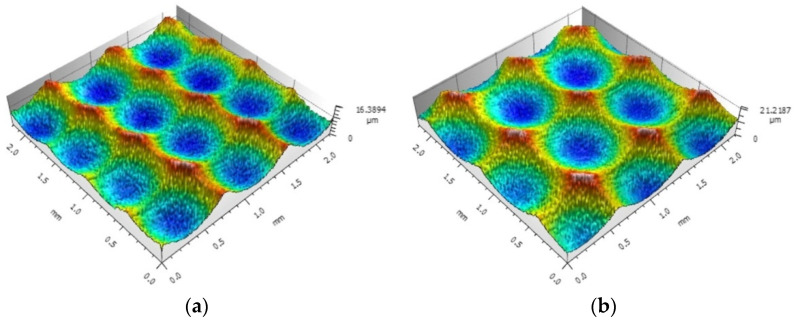
Surface topography under different *S_tr_*: (**a**) *S_tr_* = 0.308; (**b**) *S_tr_* = 0.716.

**Figure 9 materials-15-01707-f009:**
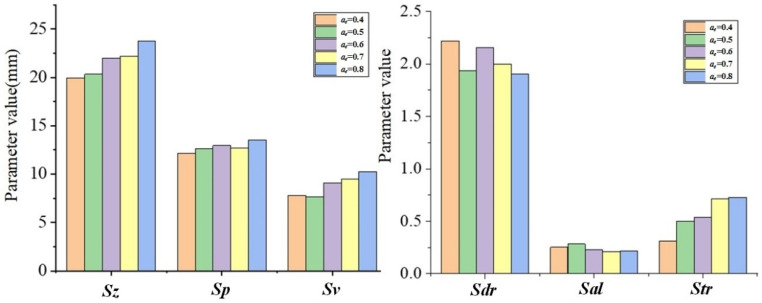
The topography parameter changes with *a_e_*.

**Figure 10 materials-15-01707-f010:**
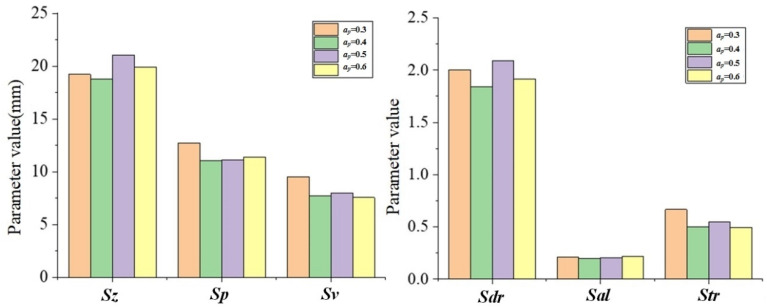
The topography parameter changes with *a_p_*.

**Figure 11 materials-15-01707-f011:**
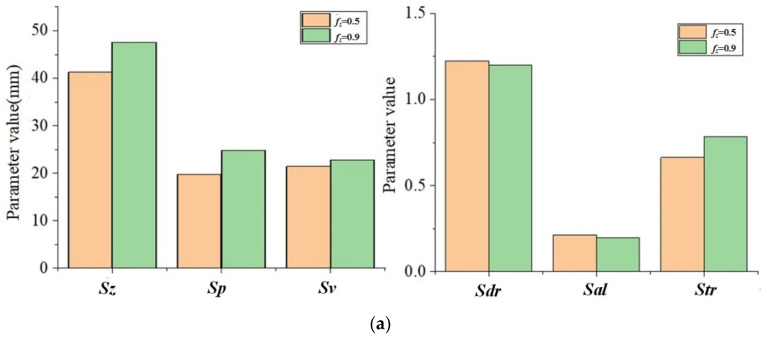
The topography parameter changes with *f_z_*: (**a**) *a_e_* = 1, *a_p_* = 0.5; (**b**) *a_e_* = 0.8, *a_p_* = 0.5.

**Figure 12 materials-15-01707-f012:**
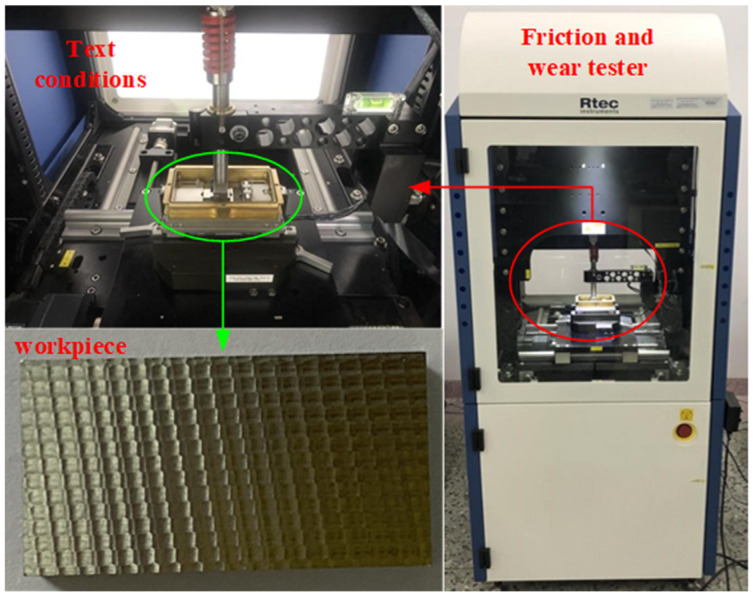
The friction and wear experiment.

**Figure 13 materials-15-01707-f013:**
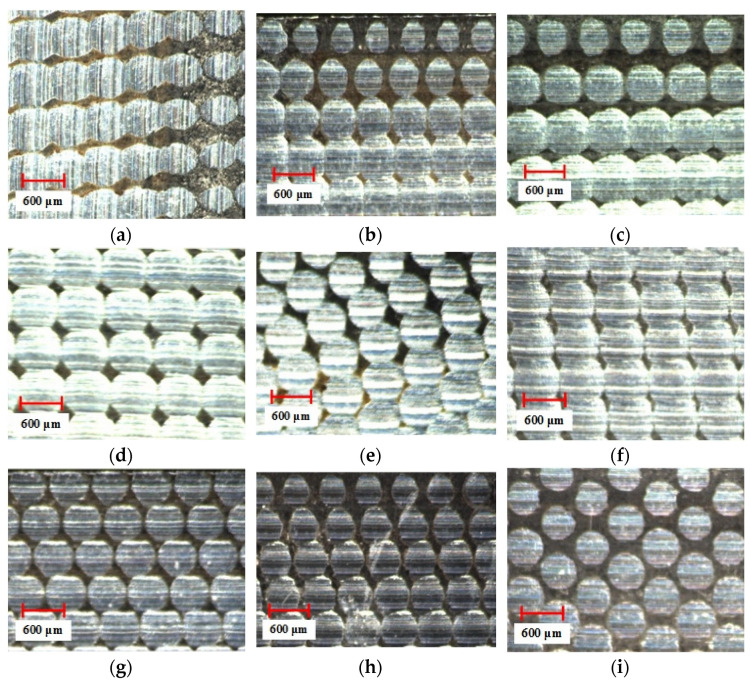
The surface topography after wear: (**a**) Group A1; (**b**) Group A2; (**c**) Group A3; (**d**) Group A4; (**e**) Group A5; (**f**) Group B2; (**g**) Group B3; (**h**) Group B4; (**i**) Group C2.

**Figure 14 materials-15-01707-f014:**
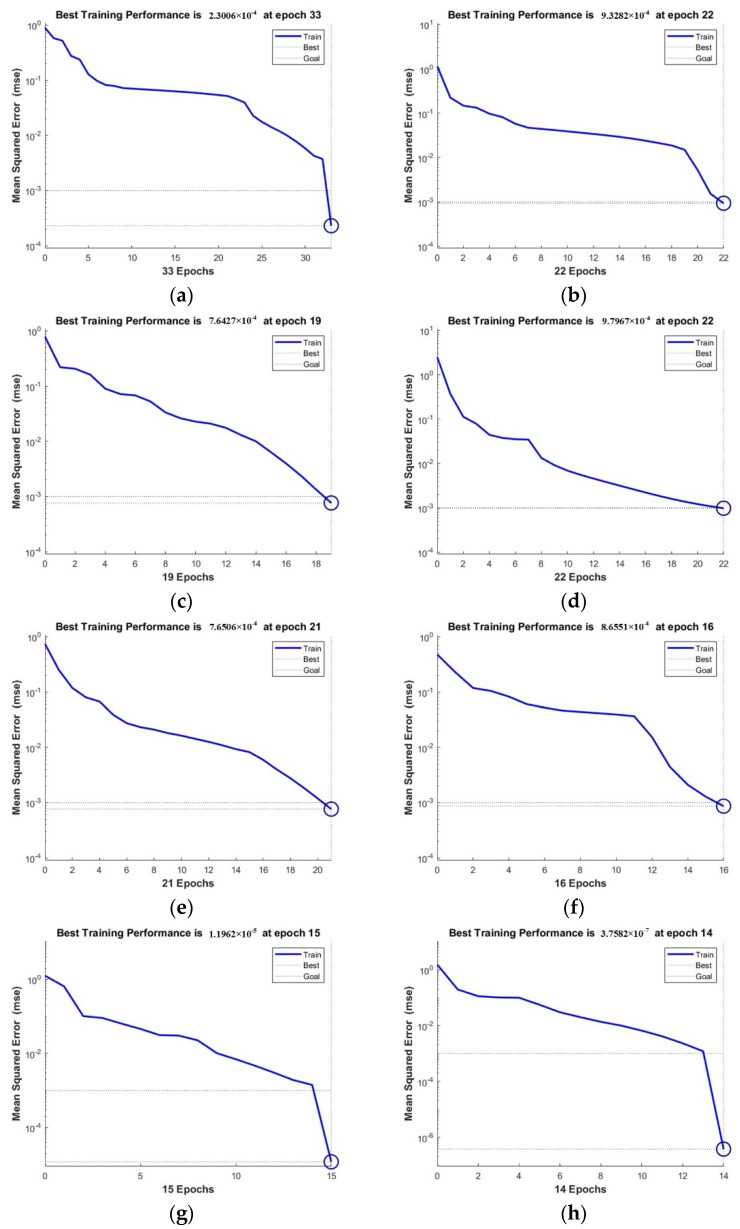
Calculation error of different hidden layer nodes: (**a**) *i* = 5; (**b**) *i* = 6; (**c**) *i* = 7; (**d**) *i* = 8; (**e**) *i* = 9; (**f**) *i* = 10; (**g**) *i* = 11; (**h**) *i* = 12.

**Figure 15 materials-15-01707-f015:**
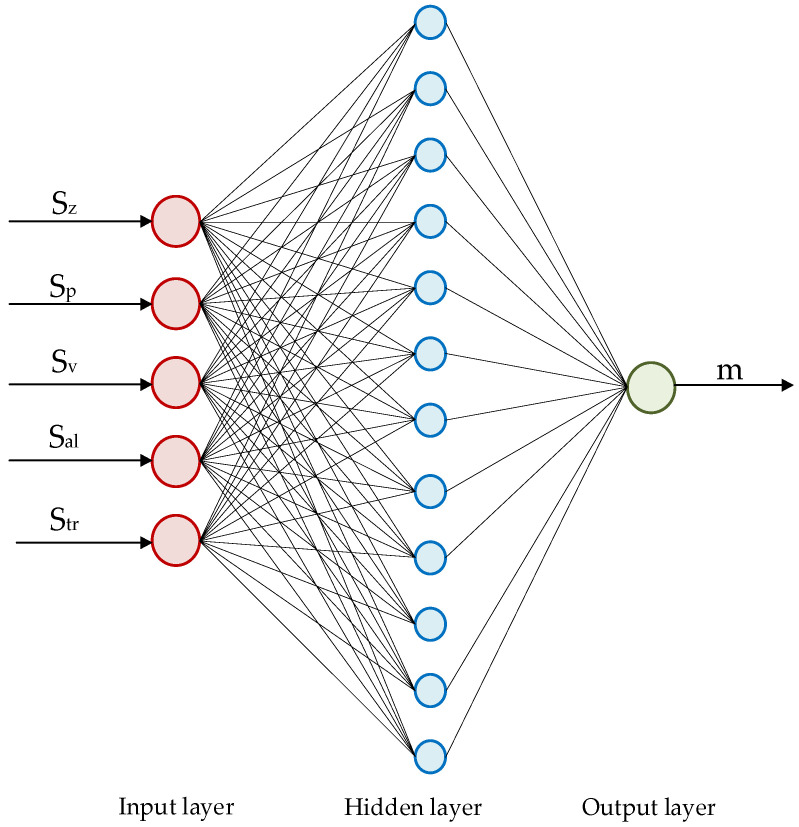
BP neural network model structure diagram.

**Figure 16 materials-15-01707-f016:**
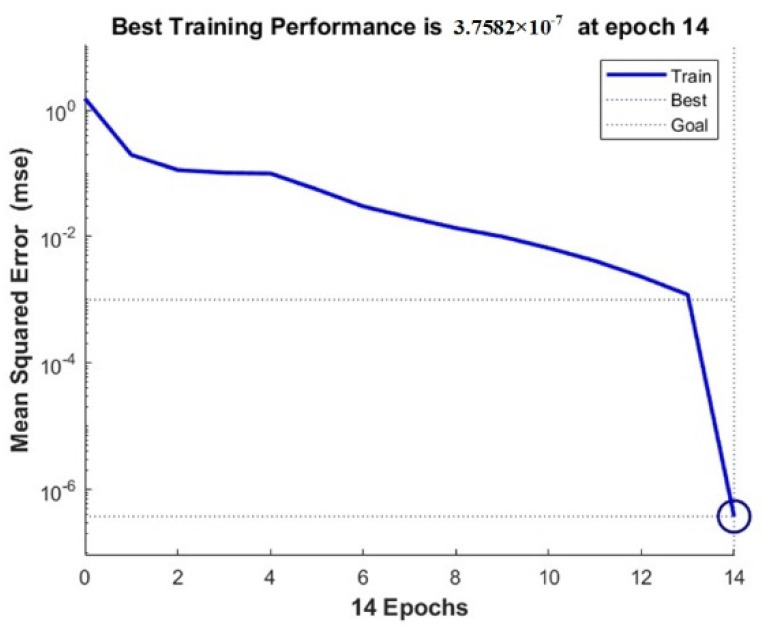
Error curve.

**Table 1 materials-15-01707-t001:** Cr12MoV Chemical composition.

Element	Cr	Ni	Cu	V	Mo	C	Mn	Si	S	P
Content (%)	11.0–12.5	≤0.25	≤0.30	0.15–0.30	0.4–0.6	1.45–1.50	≤0.4	≤0.4	≤0.03	≤0.03

**Table 2 materials-15-01707-t002:** *a_e_* single factor test.

Groups	*f_z_* (mm/z)	*a_e_* (mm)	*a_p_* (mm)	*n* (r/min)	Processing Angle
A1		0.4			
A2		0.5			
A3	0.4	0.6	0.3	10,000	30°
A4		0.7			
A5		0.8			

**Table 3 materials-15-01707-t003:** *a_p_* single factor test.

Groups	*f_z_* (mm/z)	*a_e_* (mm)	*a_p_* (mm)	*n* (r/min)	Processing Angle
B1	0.4	0.6	0.3	10,000	30°
B2			0.4		
B3			0.5		
B4			0.6		

**Table 4 materials-15-01707-t004:** *f_z_* single factor test.

Groups	*f_z_* (mm/z)	*a_e_* (mm)	*a_p_* (mm)	*n* (r/min)	Processing Angle
C1	0.4	0.4	0.3	10,000	30°
C2	0.6				
C3	0.5	1	0.5		
C4	0.9				
C5	0.5	0.8	0.5		
C6	0.8				

**Table 5 materials-15-01707-t005:** The topography parameter value of the processed surface.

Group	*S_z_* (um)	*S_p_* (um)	*S_v_* (um)	*S_al_*	*S_tr_*	*S_dr_*
A1(C1)	19.951	12.175	7.776	0.283	0.239	2.325
A2	20.319	12.646	7.673	0.252	0.308	2.218
A3	22.013	12.944	9.069	0.227	0.535	2.153
A4(B1)	22.219	12.723	9.495	0.213	0.716	2.001
A5	23.772	13.526	10.247	0.217	0.727	1.901
B2	18.763	11.038	7.725	0.195	0.503	1.638
B3	21.072	11.097	7.974	0.205	0.511	2.092
B4	19.927	12.374	7.553	0.216	0.496	1.913
C2	16.893	9.197	7.696	0.234	0.766	1.935
C3	41.256	19.793	21.463	0.214	0.663	1.225
C4	47.598	24.849	22.749	0.197	0.784	1.198
C5	40.361	23.014	17.347	0.305	0.595	1.369
C6	41.469	21.622	19.847	0.227	0.670	1.238

**Table 6 materials-15-01707-t006:** Measurement data.

Group	A1	A2	A3	A4	A5	B2	B3	B4	C2
Amount of wear (mg)	0.0044	0.0022	0.0032	0.0061	0.0051	0.003	0.0015	0.0027	0.0054
Coefficient of friction	0.5373	0.4835	0.5043	0.6188	0.5596	0.2843	0.2343	0.4776	0.5681

**Table 7 materials-15-01707-t007:** Sample and test data.

Group	*S_p_* (μm)	*S_v_* (μm)	*S_z_* (μm)	*S_al_*	*S_dr_*	*m* (mg)	*u*
A1	12.646	7.673	20.319	0.252	2.218	0.0044	0.5373
A2	12.723	9.495	22.218	0.213	2.001	0.0022	0.4835
A3	12.175	7.776	19.951	0.283	2.325	0.0032	0.5043
A4	12.944	9.069	22.013	0.227	2.153	0.0061	0.6188
A5	13.526	10.247	23.773	0.217	1.901	0.0051	0.5596
B2	11.038	7.725	18.763	0.195	1.638	0.0030	0.2843
B3	11.097	7.974	19.071	0.205	2.092	0.0015	0.2343
B4	12.374	7.553	19.927	0.216	1.913	0.0027	0.4776
C2	9.197	7.696	16.893	0.234	1.935	0.0054	0.5681

**Table 8 materials-15-01707-t008:** Normalized samples and test data.

Group	*S_p_* (μm)	*S_v_* (μm)	*S_z_* (μm)	*S_al_*	*S_dr_*	*m* (mg)	*u*
A1	0.292	−1	−1	0.295	0.688	0.128	−0.049
A2	0.354	0.415	0.880	0.590	0.056	−1.005	0.967
A3	0.532	0.084	0.786	0.272	0.499	1.004	1.000
A4	1	1	1	0.500	0.234	0.488	−0.018
A5	0.086	0.920	0.135	1	1	−0.488	−0.581
B2	−1	0.959	0.403	−1	−1	−0.590	−0.999
B3	0.952	0.766	0.027	0.772	0.321	0.241	−0.634
B4	0.074	1.093	0.047	0.522	0.199	−0.364	0.897
C2	2.479	0.982	1.017	0.113	0.135	0.630	0.786

**Table 9 materials-15-01707-t009:** Wear prediction error.

Experimental measurements	0.0044 g	0.0061 g	0.0027 g
Model prediction	0.0039 g	0.0051 g	0.0033 g
Error percentage	11.36%	16.39%	14.81%

**Table 10 materials-15-01707-t010:** Friction coefficient prediction error.

Experimental measurements	0.5373	0.6188	0.4476
Model prediction	0.5041	0.7020	0.4812
Error percentage	6.18%	13.45%	7.51%

## Data Availability

Not applicable.

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
