# Peer review of "Analysis of High-Speed Milling Surface Topography and Prediction of Wear Resistance"

_materials, 2022, doi:10.3390/ma15051707_

Round 1

Reviewer 1 Report

In this work, the authors investigated the high-speed milling surface topography and presented prediction model related to wear resistance. Through the experiment, the effect of manufacturing parameters was clearly investigated and based on the machine learning, the wear prediction and friction coefficient prediction were conducted. The result and analysis were clear and understandable, however, to improve the readability, the minor revision is required before accept this work.

  1. In this work, there are various parameters, fz, ae, and ap that can change surface topography and resultant values, Sz, Sp, Sv, Sal, Str, and Sdr. For resultant values, I could understand what they mean. However, for manufacturing parameters (fz, ae, and ap), it is hard to intrinsically find what they represent. To improve readability for readers, please clearly indicate the parameters and what they represent.

  1. From Figure 9 to Figure 11, the authors summarized the changes in topography with various ae, ap, and fz. The effect of each parameter was clearly investigated and explained. However, as I mentioned in comment 1, the authors should consider the readability. I know the detail value of parameters are organized in Table 2, 3, and 4. But, to make more understandable graph, I recommend to indicate each main parameter values on graph. (for example, ae values on Figure 9)

  1. In Figure 14 and relevant content, it looks like the as increase the number of hidden layer nodes (i), the error is constantly decreased. Could increasing the number of hidden layer nodes further reduce the error? Furthermore, does it take the same amount of time for training the model regardless of whether the number of hidden layer nodes increase or not?

Author Response

Dear Editor:

We appreciate the thorough reviews and positive comments provided by both yourself and the reviewers on our submitted paper " Analysis of High-speed Milling Surface Topography and Pre-diction of Wear Resistance " (materials-1572518). We appreciate your solid expertise. According to the modified comments, we made careful changes to the paper (Modified part plus blue). Detailed replies to the questions are presented as follows:

Comment No.1:

In this work, there are various parameters, fz, ae, and ap that can change surface topography and resultant values, Sz, Sp, Sv, Sal, Str, and Sdr. For resultant values, I could understand what they mean. However, for manufacturing parameters (fz, ae, and ap), it is hard to intrinsically find what they represent. To improve readability for readers, please clearly indicate the parameters and what they represent.

Response to comment No.1:

Thank you very much for your suggestion, I totally agree with your point of view. We have made a supplementary introduction to the corresponding parts of the text. First of all, we indicate the meanings of the processing parameters mentioned in the text in the corresponding tables and figures. In addition, the Nomenclature section is given at the end of the article to explain the meaning of these parameters.

Comment No.2:

From Figure 9 to Figure 11, the authors summarized the changes in topography with various ae, ap, and fz. The effect of each parameter was clearly investigated and explained. However, as I mentioned in comment 1, the authors should consider the readability. I know the detail value of parameters are organized in Table 2, 3, and 4. But, to make more understandable graph, I recommend to indicate each main parameter values on graph. (for example, ae values on Figure 9).

Response to comment No.2:

Thank you very much for your suggestion, I still very much agree with your point of view. We have made changes to Figures 9 through 11 in the text, indicating the value of each parameter in the figures.

Comment No.3:

In Figure 14 and relevant content, it looks like the as increase the number of hidden layer nodes (i), the error is constantly decreased. Could increasing the number of hidden layer nodes further reduce the error? Furthermore, does it take the same amount of time for training the model regardless of whether the number of hidden layer nodes increase or not?

Response to comment No.3:

In this paper, the number of hidden layer nodes was determined according to formula (17). We calculated that the value interval of the hidden layer node was [4, 12] according to formula (17). Therefore, the maximum value of hidden layer nodes in this paper is 12.

If the number of hidden layer nodes increases, the computation time to train the model will also increase.

Best wishes.

Reviewer 2 Report

Manuscript Title: Analysis of High-speed Milling Surface Topography and Prediction of Wear Resistance

Comments for the author, which must be addressed in the revised version.

  1. Add quantitative results in the abstract.
  2. Improve the flow in the introduction.
  3. Define the assumptions in the section 2 and add the references of the equations if chosen from previous published results.
  4. In section 3.1, mention the material of the milling cutter.
  5. Why is the single factor experimental technique selected over DoE?
  6. Line 272, define the Figure number.
  7. In section 5.1.1, how the input parameters of the tribo-tester?

Author Response

Dear Editor:

We appreciate the thorough reviews and positive comments provided by both yourself and the reviewers on our submitted paper " Analysis of High-speed Milling Surface Topography and Pre-diction of Wear Resistance " (materials-1572518). We appreciate your solid expertise. According to the modified comments, we made careful changes to the paper (Modified part plus blue). Detailed replies to the questions were presented as follows:

Response to comment No.1:

Thank you very much for your suggestion. We have made changes to the Abstract section of this paper, mainly by adding the resulting errors in wear resistance predictions to the abstract. Modifications have been highlighted in blue.

Response to comment No.2:

Thank you for your suggestion. We have rephrased the introductory part of this paper. First of all, the overall logical relationship of the introduction was sorted out, and the unsmooth connection was adjusted; Secondly, the opening part of the introduction was re-expressed, indicating the background of this study, and deleting redundant content; Finally, a more concise summary of the existing research was made, and a more accurate summary of the research content and purpose of this paper was re-introduced.

Response to comment No.3:

Thank you for your suggestion, I totally agree with your point of view. Since some parameters were not clearly defined, it affected the readability of this section. We have defined the parameters not specified in Chapter 2 and marked them in blue. In addition, the equations in Chapter 2 were respectively derived from the geometric relationships in Figure 1 and Figure 2. In order to make the formulation of the equation clearer, we explained its parameters at the beginning of Chapter 2.

Response to comment No.4:

Thank you very much for your suggestion, I still very much agree with your point of view. We have supplemented the material of the milling cutter in the corresponding section of the article and marked it in blue. The base material of the milling cutter insert used in the milling experiment in this paper is cemented carbide. The blade surface is coated with JC6102 and its hardness is around 70 HRC.

Response to comment No.5:

Thank you very much for your suggestions. The methods you mentioned had a very good effect in many experiments. The single factor experiment method was mainly used to study the influence of a single variable on the result. This paper was to explore the effects of fz, ae and ap on the surface topography and topography parameters respectively. Therefore, the single factor experiment method was used for milling experiments.

Response to comment No.6:

Thank you very much for your suggestions. We have added the figure number there and marked it in blue.

Response to comment No.7:

The friction and wear experimental parameters in this section are as follows: the sliding frequency is 2Hz, the single stroke is 20mm, the applied load is 80N, and the test time for each group of specimens is 6min. We have supplemented this part to the article and marked it in blue.

Best wishes.
